# The Role of Personalization in the User Experience, Preferences and Engagement with Virtual Reality Environments for Relaxation

**DOI:** 10.3390/ijerph19127237

**Published:** 2022-06-13

**Authors:** Susanna Pardini, Silvia Gabrielli, Marco Dianti, Caterina Novara, Gesualdo M. Zucco, Ornella Mich, Stefano Forti

**Affiliations:** 1Department of General Psychology, University of Padova, Via Venezia 8, 35131 Padova, Italy; caterina.novara@unipd.it; 2Digital Health Lab, Centre for Health and Wellbeing, Fondazione Bruno Kessler, Via Sommarive 18, 38123 Trento, Italy; sgabrielli@fbk.eu (S.G.); dianti@fbk.eu (M.D.); forti@fbk.eu (S.F.); 3Department of Philosophy, Sociology, Education and Applied Psychology (FISPPA), University of Padova, Palazzo del Capitanio, Piazza Capitaniato 3, 35139 Padova, Italy; zucco@unipd.it; 4Center for Information and Communication Technology, Fondazione Bruno Kessler, Via Sommarive 18, 38123 Trento, Italy; mich@fbk.eu

**Keywords:** virtual reality, personalized virtual environment, user-experience, mental well-being, non-clinical population, mixed-method study

## Abstract

Virtual Reality Environments (VREs) are widely deployed in mental health treatments, often associated with relaxation techniques. The personalization of natural VR-based scenarios is a key element that can further facilitate users’ sense of presence and relaxation. This study explored the role of VREs’ personalization in the user experience with an environment supporting relaxation, by deploying mixed methods. Methods: A non-clinical sample of 20 individuals participated in exposure to a supportive body-scan-guided relaxation VRE. In the personalized conditions, the participants had the option of choosing the context (e.g., sea, mountain, or countryside) and including in the scenario different types of sounds, visual elements, and changing the time of day and weather. In the standard conditions, individuals were exposed to a relaxing VRE, but they could choose neither the context nor the auditory and visual elements. The order of presentation of the personalized vs non-personalized environments was randomized. Measures regarding relaxation, state-anxiety perceived levels, VRE-related symptoms, the usability of the Virtual Reality (VR) setting, sense of presence, pleasure, activation, engagement, and level of immersion experienced were collected before and after exposure to the VR environments. Results: Findings showed that personalized VREs were preferred by users. Participants generally preferred to experience a greater immersivity, pleasure, engagement, and relaxation in the personalized virtual settings. Conclusion: The study further confirms the role of personalization as a component positively contributing to relaxation and engagement. Future research may further assess this effect in the context of large-scale controlled studies involving clinical and non-clinical populations.

## 1. Introduction

State-anxiety could be defined as a cognitive, behavioral, and physiological reaction to specific situations perceived as potentially dangerous [1,2]; it changes over time and can be coped with by using different intervention strategies, such as specific relaxation training programs.

Techniques and protocols that allow relaxation are helpful in promoting a status of general well-being, improving mental health, playing an important role in the prevention of cardiovascular diseases, weight, joint, and diabetic disorders, and reducing anxiety as well [3,4]. Several relaxation programs, such as the Progressive Muscle Relaxation Technique (PMRT), body scan, deep breathing, and meditation, besides being cost-effective and easy to learn, effectively manage anxiety, depression, and physiological activation, e.g., [3,4,5]. For example, a recent study [6] showed that PMRT and deep breathing improved the state of relaxation when the participants in the intervention group were compared to those in a control group. The Body Scan technique is based on progressively and gradually learning to pay attention to the different body areas. This practice has a relevant impact on promoting somatic awareness and managing distress [7], with prompt, positive consequences on psychological well-being [8]. Moreover, the body scan is a shorter and lower-cost technique if compared to a complete mindfulness program, as well as easier to administer [9].

Considering the impact of the COVID-19 pandemic on increasing stress and anxiety symptomatology in the general population, digital psychological interventions, accessible remotely and in a self-administered mode, should be implemented and validated [10].

A recent review analyzed the scientific evidence on the feasibility and effectiveness of Virtual Reality (VR) in the general population, showing VR as an effective solution in promoting relaxation [11]; notably, pleasant, and immersive VR environments have been shown to improve the ability to cope with distress, and enhance relaxation and positive mood [12,13,14]. Additionally, virtual reality applied to mindfulness and body scan has been recently investigated, showing positive effects on relaxation [15,16,17,18]. VR scenarios typically use audio and visual elements, representing natural stimuli that facilitate stress recovery and contribute to psychological and physiological relaxation [19,20]. Specifically, exposure to blue and green natural environments promotes well-being due to physical and mental reactions to these stimuli [21,22]. Since relaxation exercising in a natural context may be less practical to realize in several circumstances and by different target user groups (e.g., older adults), providing VR natural environments supporting these experiences is of particular interest and should promote well-being [23,24]. Recent findings have shown no key differences between a real or virtual natural environment in eliciting a sense of presence [25,26], confirming the potential benefits of enabling relaxation in personalized environments.

In general, VR is widely used in the treatments of various mental disease [27], often in association with relaxation training and techniques (e.g., body scan, muscle relaxation exercises). It favors multi-sensory stimulation, sense of presence, and attaining a relaxed state [28,29]. Data suggest that VR can facilitate the learning of relaxation techniques through VR exposure to natural environments that are intrinsically relaxing [18].

Evidence shows that the contribution of personalized scenarios further promote relaxation. These VR contexts, developed by deploying user-centered design methods [28], enable the participants to more realistically experience emotional conditions in a virtual setting, increasing relaxation, sense of presence, and security [30,31]. Personalized VR scenarios are characterized by particular multi-sensorial content and can be tailored to users’ relevant preferences and needs [28,32,33]. Exposure to a personalized environment enhances the users’ involvement and their sense of presence [30,31,34,35,36] by exploiting VR’s capacity to remove many of the distracting factors in the natural world and, consequently, facilitating immersion and relaxation [37]. These digital interventions may also help reduce the costs of standard psychotherapy treatments, in addition to being increasingly scalable and able to lower costs [38,39,40]. They help to standardize and offer remote access to psychological and medical treatments, as required by the current COVID-19 pandemic context [41]. A recent study proposed a personalized VR model for relaxation, based on critical variables (including the personalities and emotional states of individuals) that were modelled and controlled by an algorithm used to customize the VR environment according to users’ needs and profiles [37].

However, to the best of our knowledge, few studies have explored the role of personalization of VR-based scenarios on the user experience [37], and the effect of the integration of relaxation training in VREs. One of the most recent studies in this direction [18] showed that being exposed to a virtual environment for relaxation, including a body-scan-guiding audio track, should help improve the participants’ relaxation compared to VREs including a breathing control audio track.

Based on this information, we assume that the users’ perceptions, derived from the exposure to the features of the VR system, need to be more deeply assessed as a starting point to then investigate its effectiveness in more detail. A systematic investigation of the users’ experience and usability are essential to understand and improve the deployment of hardware components (e.g., the head-mounted display) and of the virtual setting design [42] in future VR-based digital interventions. The assessment of subjective experience and usability comprises a wide variety of elements that need to be adapted, based on the VRE intended use [42]. First, *motion-simulator sickness* must be assessed as a component that can affect the VR experience and could comprise headaches, salivation, eyestrain, nausea, and a general state of discomfort. The assessment of some aspects concerning the *quality of VR graphics* (e.g., resolution, movement of visual elements, shapes, color contrast), *physical performance,* and *synchronization* (that in our relaxing VRE should be understood as the time delay between the user’s movement into the VRE and what it is expected to see), as well as *user interface* layouts are important elements that could influence, for example, immersivity and sense of presence constructs. A *heuristic assessment* helps obtain direct users’ feedback and could include the investigation of natural engagement, realistic feedback, orientation support, and sense of presence [43,44]. Stanney et al. [45] suggested using the Multi-criteria Assessment of Usability for Virtual Environments (MAUVE) to systematically evaluate users’ experience and usability in the VRE context. It comprises ten criteria divided into two main categories. The *VE System Interface* included *Interaction features,* such as Wayfinding (the possibility for the user to know their position and direction in the environment), Object Selection, Manipulation, and Navigation in the VRE, and the *Multimodal System Output* that includes Visual, Auditory, and Haptic features. The *VE User Interface* is, in turn, divided into *Engagement components* (presence and immersivity), and *Side Effects* (comfort, sickness, after-effects).

The study of user experience related to the role of personalized VR scenarios has attracted interest in the scientific community although, as mentioned before, the studies published are more on a theoretical level, e.g., [37]. To further understand the role of the personalization component in VR scenarios, the main aim of our pilot and exploratory study was to investigate the experience of users to obtain preliminary data about preferences, pleasure, satisfaction, engagement, immersivity, sense of presence, and the subjective perception of relaxation and state-anxiety whether being exposed to a standard or a personalized relaxing VRE. Inspired by the user experience’s background described above, and to generate integrated support from qualitative and quantitative data, we obtained information about the physical discomfort, related to the heaviness or encumbrance of the head-mounted display, the simulator-sickness, the wayfinding, and the experience of the auditory and visual elements, both in the standard and personalized scenarios. An explicit emphasis was placed on engagement, immersivity, sense of presence, the subjective perception of relaxation and state-anxiety levels, and pleasure. The study design was based on a mixed-methods approach inspired by the Obesity-Related Behavioral Intervention Trials (ORBIT) framework [46] for design (Phase Ib) of digital interventions and their preliminary testing (Phase IIa). In this phase, a proof-of-concept implementation of the personalized VR scenario was realized, and preliminary testing was completed to investigate level of engagement and the user experience with a convenience sample of non-clinical individuals. In this context, it is useful to apply a within-subjects design where users act as their controls in a pre–post comparison. The sample size can be small because the focus is mainly on the user experience outcome, and sample size estimation is not required. Moreover, the sample can be selected from accessible individuals since this phase will aim to understand if the intervention deserves an increased depth of analysis, improvement, and future testing. We hypothesized that: (1) relaxing VR scenarios promote relaxation and reduce state-anxiety symptoms; (2) the possibility of personalizing the VR environment can be perceived as more engaging for users. By following a person-centered approach to enhancing the effect of standard interventions [28], we present below the main outcomes of our design and preliminary testing phases.

## 2. Methods and Materials

### 2.1. The Virtual Environment Design

The virtual environment design was based on some hardware equipment tools composed of an “Alienware m15 Ryzen Edition R5” workstation (CPU 5900HX, 32Gb ram, 1Tb SSD RTX3070 8Gb), a link cable (PC VR; USB 3 Type-C–5m), and an Oculus Quest 2^®^ head-mounted display related to the workstation. The Oculus device was chosen because it is less expensive and easier to use than other devices. This is in line with our mission to extend the use of this setting for psychological treatment in everyday life. The software equipment was composed of a series of 360° natural environments that were characterized by auditory stimuli drawn from the *freesound* database (https://freesound.org/), and the visual stimuli were developed based on Unity^®^, Polyhaven, and HDRIhaven. The VR environment was developed by the Digital Health Lab at Fondazione Bruno Kessler (FBK) Research Center (Trento, Italy). Specifically, users may choose to experience relaxation, based on their preference, in one of three realistic scenarios, a mountain, a marine, and a countryside environment, that may be selected and personalized with the support of a technical operator.

For each selected environment, the participants had the option to personalize a series of sub-categories of the elements, including different types of sounds (e.g., animal sounds, wind rustle, type of music), visual elements (e.g., the presence of people, objects, or animals), the position in the virtual space, the presence of people, and to change the time of day and weather conditions (an example is shown in Figure 1). The operator had the possibility of customizing each scenario based on a dashboard interface connected to Oculus via a cable. Specifically, the interface presented a series of icons related to the different customizable variables (such as music, wind, weather conditions, time of day, presence of people) (Figure 1). The connection between the PC and the viewer allowed the operator to see what the participant observed during the experience.

The natural VR scenario was integrated with a body scan audio track provided by the Oculus that focused on the different parts of the body and sensations experienced in a gradual sequence. The audio track was inspired and adapted from the body scan exercise as part of the Mindfulness-Based Stress Reduction (MBSR) program and the PMRT [47,48,49,50]. The track lasted 7 min. Body scan was deployed in the current research by following the results in the Pizzoli study [18] that showed that this technique had a more effective impact on relaxation than breathing exercises.

### 2.2. Procedure

All of the sessions took place at the Center for Health and Wellbeing–Fondazione Bruno Kessler, Trento (Italy). The first pilot study, based on a convenience sample of individuals from a non-clinical population, was realized between 10 August 2021 and 25 September 2021. The recruitment phase was based on a snowball sampling method, and it was carried out via email.

A total of 27 individuals agreed to participate in the study. After applying the exclusion criteria, 20 were included in the analysis. The exclusion criteria were: (i) participant had a mental disorder diagnosis; (ii) a score on the STAI-Y2 indicating moderate or severe trait-anxiety levels (raw cut point > 50; [51,52]); (iii) a score > 9 on the DASS-21 depression subscale; (iv) a score > 6 on the DASS-21 anxiety scale; (v) a score > 10 on the DASS stress scale [53,54]. Seven of the participants were excluded to control the effect that the mental disorders’ diagnosis, trait-anxiety, and depressive symptomatology may have when comparing state-anxiety before and after VREs’ usage.

In phase T0, individuals signed the informed consent to participate in the research. They also filled out a socio-demographic survey and a series of self-report scales to investigate distress, depressive and anxiety symptoms, the perceived relaxation level, and emotional states. Then, the participants were invited to wear the head-mounted display to be exposed to a short body-scan-guided relaxation session either with a default-standard VR condition or a customized VREs (personalized VR condition). The order of presentation of each type of VR condition was randomized, and a mixed-methods approach was deployed to collect data. The customized VR experience was personalized according to the participant’s preferences on the type of environment, music background, meteorological conditions, presence of objects or other participants, and daytime (first administration phase).

Then, in phase T1, the individuals compiled the instruments administered during phase T0 to measure their state of anxiety and emotions, in addition to an ad-hoc questionnaire deployed to collect evidence on usability, sense of presence, immersion, engagement, and VRE-related symptoms. Afterwards, they were exposed to a second VR experience in the other condition of the study.

During phase T2, the participants filled out the same self-report tools of phase T1. Then, the researchers asked the participants what kinds of VR conditions they preferred and if they had suggestions for improvements.

The assessment procedure took about 45–50 min to be completed, the VRE personalization took approximately 10–15 min, and the body scan session lasted about 7 min. All of the participants knew they could ask to interrupt the experience at any stage.

In Figure 2, a brief description of the procedure is presented.

### 2.3. Materials

To investigate the study’s main constructs, we relied on the following self-report questionnaires and open-ended questions, typically employed in the measurement of VR scenarios, to collect a combination of quantitative and qualitative data.

The *Socio-demographic survey* consisted of the following variables: gender, age, nationality, mother tongue, level of education, marital status, employment status, type of medical disorder/illness/disease (such as some neuromuscular disorders), or a psychological problem previously diagnosed. Participants were also asked to report if they were taking drugs, had already experienced relaxation activities to manage anxiety in the past, or were familiar with the use of virtual reality devices.

The ad hoc *questionnaire for Virtual Reality* investigating different aspects of Virtual Reality scenarios and User Experience was composed of the following sections: (1) The Virtual Reality Symptom Questionnaire (VRSQ) [55,56] is a dichotomous measure aiming to assess symptoms of cybersickness. Eight items investigate general physical side effects, such as fatigue, headache, nausea, concentration difficulties, and the other five are about visual effects, for example, blurred vision and tired eyes; (2) The 36 items, based on a 7-point Likert scale, were inspired by the Presence Questionnaire and the Immersive Tendencies Questionnaire [57] and offered a qualitative set of information. The items investigating aspects, such as the realism of the environments (13 items) (e.g., “How natural did your interactions with the environment seem?”), engagement (three items) (e.g., “I felt involved by the visual elements that surrounded me within the virtual environment”), immersiveness (nine items) (e.g., “How immersed were you in the virtual environment experience?”), the tools’ usability and quality of the interface (six items) (e.g., “How much did the control devices interfere with the performance of assigned tasks or with other activities?”), and emotional states (eight items) (e.g., “During the experience, I felt tense”), were modified to be adapted to the virtual reality environment; (3) At the end of the entire procedure, an open-ended question assessed if the participant preferred the standard or the personalized VRE. Cronbach’s alpha for each sub-group was calculated (0.69 < Cronbach’s alpha < 0.88); 4) We administered the ad hoc satisfaction questionnaire to assess the usefulness of the personalized VRE (pVRE) body scan procedure; specifically, based on a dichotomous question, the participants were asked if they considered the body scan audio track helpful or not in promoting relaxation, and two open-ended questions were asked to investigate criticisms or suggestions for improvements.

The *Depression, Anxiety and Stress Scale-21* (DASS-21) [53,54] is a self-report questionnaire consisting of 21 items, seven items per subscale, investigating depression, anxiety, and stress. Items are answered according to the presence and intensity of symptoms in the last week on a 4-point Likert scale (0–3). Based on the original version, DASS demonstrated a good internal consistency. In addition, psychometric analyses of the Italian samples showed that the questionnaire investigated general distress plus three additional orthogonal dimensions (anxiety, depression, and stress) and demonstrated good internal consistency, temporal stability, and criterion-oriented, convergent, and divergent validity. The DASS-21 reliability in this study population was adequate, with a Cronbach’s alpha of 0.90 for the total score, 0.78 for the depression scale, 0.71 for the anxiety scale, and 0.90 for the stress scale.

The *State-Trait Anxiety Inventory-Y* (STAI-Y) [51,52] is a self-report questionnaire consisting of two 20-item scales investigating state- and trait-anxiety. State-Anxiety could be defined as a transitory anxiety reaction to an event, or a condition perceived as adverse, characterized by feelings of tension, apprehension, nervousness, and worry. Trait-Anxiety is, instead, a more stable feature related to perceiving stressful situations, such as dangerous or threatening situations in general or as a frequency response in abnormal conditions. Each item is evaluated based on a 4-point Likert scale (1–4). The total score for both scales ranges from 20 to 80, with higher scores indicating more severe anxiety. For the Trait-Anxiety score (investigated with the STAI-Y2), Cronbach’s alpha was equal to 0.80, indicating a good internal consistency. Regarding the State-Anxiety (measured with the STAI-Y1), the Cronbach’s alpha indices for all three of the administrations ranged from 0.84 to 0.94.

The *Self-Assessment Manikin* (SAM) [58,59] is a non-verbal imagery-based assessment technique that directly measures the pleasure, arousal, and dominance associated with a person’s affective reaction to a wide variety of stimuli. The pencil-and-paper version required placing an “X” either on or between each of the five figures (resulting in a 9-point scale). The meaning of each scale is described to respondents, and they are asked to place the “X” on the figure (or between the figures) that best represents how they currently feel. Valence is depicted from positive (a smiling figure), to neutral, to negative (a frowning figure). Arousal ranges from high arousal (eyes wide open) to low arousal (eyes closed). When the same figures were used, the arousal scale also depicts the intensity of arousal with additional imagery over the abdomen area that ranges from high intensity (imagery representing an explosive-like burst) to low intensity (imagery representing a tiny pinprick). Finally, dominance/control ranges from feeling controlled or submissive (a petite figure) to feeling in control or dominant (a huge figure).

The *Visual Analogue Scale* (VAS) measured both relaxation and sense of presence before and after each VRE. Specifically, the participants had to express how relaxed they felt on a line of 10 cm length (VAS_relax: 0 = not at all relaxed; 10 = completely relaxed; VAS_sense of presence: 0 = absent; 10 = complete).

These data collection methods were deployed in parallel for a holistic and multidimensional understanding of the VR experiences. In Table 1, the data collection methods and their analysis are summarized.

## 3. Results

### 3.1. Statistical Analysis

Anonymized data were processed by using SPSS Statistics version 27 software [60].

Based on the DASS-21 and the STAI-Y distributions, the range of skewness and kurtosis was from −0.974 to 1.408, falling between −2 and +2. For this reason, data are reasonably normal [61,62]. Moreover, the Kolmogorov–Smirnov and the Shapiro–Wilk tests were carried out for the DASS-21 total score, the STAI-Y2 total score, the STAI-Y1 pre-VR exposure, the STAI-Y1 post-standard VR exposure and the STAI-Y2 post-personalized VR exposure; data were not statistically significant (*p*-values: >0.05), and this constitutes further evidence of the normal data distributions.

First, Cronbach’s alpha was performed for each self-report questionnaires’ subscales. Secondly, to explore the sociodemographic features, the frequencies, means, and standard deviations were calculated.

To control the order effect of the VR scenarios (standard VRE versus pVRE), multivariate ANOVAs were conducted considering all of the questionnaires administered at the two different time points after the standard and pVR experience.

Descriptive data about users’ experiences, preferences, and engagement were derived from the frequencies. Moreover, the participants’ responses to the open-ended question relating to the preferred VR scenario were analyzed by SP and SC by using thematic analysis and were reported as frequencies. Data were analyzed thematically following an inductive, data-driven approach, based on the procedure outlined by Braun and Clarke [63]. Data codes were generated systematically, then collated into themes and applied to the entire data set to generate frequencies.

ANOVAs repeated measures were calculated for the psychological scales assessed at different time points (T0, after the standard or personalized VRE). Paired-samples *t*-tests were performed to compare the self-report scores taken at three different time points (T0, after standard or personalized VRE) and to investigate the impact of the VRE scenarios on the psychological constructs analyzed. A *p*-value equal to or less than 0.05 was considered statistically significant.

### 3.2. Preliminary Analysis

A comparison between groups was made to control the order effect of the standard and personalized VR scenarios (Table 2). No statistical differences emerged, meaning that the two VRE scenarios’ administration order had no effect.

### 3.3. Sociodemographic Characteristics of Participants

The present sample was composed of 16 (80%) women. The mean age of the participants was 34.2 (SD = 10.6; min.: 19; max.: 58), the mean of the school years was 18.9 (SD = 3.6; min.: 11; max.: 26). Twelve (60%) of the participants were single or non-cohabitant, and eight (40%) were married or cohabitant. Nine (45%) of the participants were pursuing a Ph.D. program, one was a university student, and ten (50%) were employed. No participant had a diagnosed psychological problem. Seven (45%) of the participants had familiarity with relaxation techniques to manage anxiety and stress symptoms, and six (30%) had experienced VR in the past for entertainment purposes. Means and standard deviations of total and subscales self-report scales are described in Table 3.

### 3.4. Qualitative Results

In the next sub-paragraphs, information is provided about the hardware- and software-related variables characterizing the VR experiences.

#### 3.4.1. Is the VRE Preferable with or without the Body Scan Audio Track?

The participants’ reports regarding the VR experiences and the body scan audio track showed that the body scan was considered helpful in promoting relaxation by 18 (90%) of the participants. Two individuals considered this track useful if administered alone; they reported having difficulty concentrating on the environment and the audio track content simultaneously and expressed interest in trying the experience with the audio track by imagining the relaxing environment.

#### 3.4.2. Usability, Physical Discomfort in Wearing the Head-Mounted Display, and Simulator Sickness

All of the users affirmed that Oculus was very easy to wear and use, since technical support was provided by the experimenter during the session. Six (30%) individuals recommended improving the Oculus wearability to allow greater grip and more comfort with it. In the suggestion section, three of them stated that they would prefer to choose the position in which to stay during the activity (e.g., lying down), also to manage the feeling of heaviness and encumbrance associated with the head-mounted display. Regarding the assessment of the cybersickness symptoms, two (10%) of the participants referred to mild headaches and burning eyes experienced towards the end of the VRE session with the Oculus exposure.

#### 3.4.3. Graphics Quality, Synchronization, and Wayfinding VRE-Related

Some information on the graphic quality was obtained about the perception of the audio and visual elements. All of the participants reported that audio stimuli, such as animal sounds, wind rustle, and wave movement, felt good and were consistent with the scenario. As for the visual elements, all of the individuals appreciated the colors and shapes of the objects that made up the context, evaluating them as very or totally engaging and very similar to reality. All of the individuals felt able to actively explore the environment as they decided to move their head and gaze into the virtual context. Moreover, the way they moved around the VR scenarios and the synchronization between their movement and what they saw seemed very (15 participants; 75%) or extremely (five participants; 25%) realistic to them. Moreover, users expressed a need to know their position and direction in the VREs.

#### 3.4.4. Realism of the Virtual Environments

Regarding the level of realism of the auditory and visual elements offered by the VREs, nine (45%) individuals considered the animal sounds very similar to reality (e.g., the sound of birds), 10 (50%) found the meteorological sounds extremely realistic (such as the wind and raindrops), and nine (45%) of the participants appreciated the way that the visual effects related to the meteorological conditions.

Seven (35%) individuals evaluated the placement of the visual elements in the environment very similar to a real context. Six (30%) of the participants affirmed that the colors, lights, and reflections at the transition from one moment to another of the day were very similar to real stimuli, and nine (45%) considered the movement of birds and clouds to be realistic.

Moreover, most of the participants (16; 80%) did not appreciate the presence of humans in the VR context. They instead preferred to imagine the presence of someone close to them during the VR experience. Finally, five (25%) users specified that the perception of the synchronous movement of the leaves and flowers of the countryside was artificial, suggesting allowing a less uniform fluctuation.

#### 3.4.5. Preferred VR Environment between Standard and Personalized

Eighteen (90%) of the individuals preferred the personalized environment. All of the participants expressed a subjective point of view based on their preference.

Overall, four main themes were identified as the most influential factors affecting the participants’ choices in the VREs for relaxation (Table 4): (1) the correspondence with the environment they would have chosen in a real context (relaxing as in reality); (2) the reminiscence; (3) the possibility to choose and control the elements of the VRE (control); (4) the realism of the stimuli.

Of these 18 users, 16 (89%) said it represented the real context in which they would choose to relax, and they found the possibility of personalizing the elements very important, to make the VRE more alike not only the preferred context but also the place in the real world where they would go to relax. Three (17%) individuals reported that the most important thing related to the personalization was the perception of controlling and deciding the setting, which was felt to be very impactful on their motivation to practice VR relaxation experiences. Four (22%) of the participants reported that the chosen VRE and the way they could personalize it reminded them of the places they experienced in their childhood. Even though they might have found another virtual setting more realistic, they chose the one in which they would relax better as, in their opinion, this had a significant impact on the possibility of experiencing a subjectively adequate level of relaxation. Finally, two (10%) individuals preferred the standard VR environment saying that by finding it more realistic, they felt much more comfortable in that setting.

### 3.5. Differences between Self-Report Administration after Standard and Personalized VR Experiences about Immersivity, Sense of Presence, Realism, Engagement, Usability, Subjective Arousal Perception, Sense of Relaxation, and Pleasantness

Even if the order effect of exposure to the two VR scenarios was controlled, a within-subject design was limited by the possibility that the participants realized the aim of the research, affecting their responses. In this regard, the following analyses were conducted to inform our design, by considering the subjective experience that the participants had when they were exposed to the standard and personalized VREs.

Comparisons: using Multivariate ANOVA, between the assessments conducted after the standard and personalized VRE, showed a difference for the Immersivity scale (F(1,19) = 11.33; *p* > 0.01; η2 = 0.37) with a higher sense of immersivity experienced after the personalized VR condition compared to the standard one. No differences emerged for the Usability (F(1,19) = 0.10; *p* > 0.05), the VAS_Sense of Presence (F(1,19) = 2.75; *p* > 0.05), the Realism (F(1,19) = 3.54; *p* > 0.05), the Engagement (F(1,19) = 1.06; *p* > 0.05), and the Emotional State scales (F(1,19) = 0.57; *p* > 0.05).

Repeated Measures: ANOVA, conducted to compare the three assessment phases, showed statistically significant differences between the three assessment time points (T0-T1-T2) for the STAI-Y1 (F(2,18) = 12.21; *p* < 0.001; η2 = 0.58), the SAM-Valence (F(2,18) = 8.49; *p* < 0.05; η2 = 0.49), the SAM-Arousal (F(2,18) = 4.21; *p* < 0.05; η2 = 0.25), and the VAS-relax (F(2,18) = 50.56; *p* < 0.001; η2 = 0.85). No differences emerged for the SAM-Dominance (F(2,18) = 1.47; *p* > 0.05).

Paired-samples *t*-tests were conducted as a post-hoc analysis to understand the origins of the differences between the three assessment time points. As shown from the STAI-Y1 and the VAS relax scale (Table 5), state-anxiety symptoms were higher before both the standard and personalized VR experiences. Indeed, individuals expressed, on average, a lower feeling of relaxation and more intense activation and anxiety levels before the exposure to both of the VR scenarios (t STAI (T0)/STAI (st) = 3.20; *p* < 0.01; t STAI (T0)/STAI (p) = 4.97; *p* < 0.001; t VAS_rel (T0)/VAS_rel (st) = −6.25; *p* < 0.001; t VAS_rel (T0)/VAS_rel (p) = −2.52; t SAM_A (T0)/SAM_A (p) = −2.46; *p* < 0.05). Moreover, higher relaxation levels were experienced after the personalized VR experience rather than the standard one (t STAI (st)/STAI (p) = 2.20; *p* < 0.05; t VAS_rel (st)/VAS_rel (p) = −9.95; *p* < 0.001). Regarding the Valence of emotions, the participants referred to having experienced a greater degree of pleasantness after the personalized VR setting (t SAM_V (T0)/SAM_V (p) = 3.80; *p* < 0.01; t SAM_V (st)/SAM_V (p) = 2.97; *p* < 0.01).

## 4. Discussion

The main goal of this study was to understand the experience of users to obtain preliminary data about usability, preferences, pleasure, satisfaction, engagement, immersivity, sense of presence, and the subjective perception of relaxation and state-anxiety whether being exposed to a standard or a personalized relaxing VREs, concurrently with a body scan audio track.

Experiencing the VR scenario with the audio track was appreciated by most of the participants who considered this complementary approach a useful strategy for focusing their attention on the activity and promoting relaxation. This feature deserves to be deepened with further studies that should include other conditions in which users experience either the audio track or the VR scenario alone.

About the device’s usability, wearing the head-mounted display interfered to a limited extent with exposure to the VRE. To inform our future work, we will consider some of the participants’ recommendations regarding the possibility of improving the Oculus fit by adopting a special support for the head-mounted display that balances the distribution of the device’s perceived weight, reducing interference.

Important self-reported suggestions were also obtained about the synchronization and wayfinding of the VRE, as well as the auditory stimuli, that contribute to render visual stimuli, and the experience in general, more realistic. An interesting finding was related to the presence of virtual people in the VRE. Although the representation of people was not generally appreciated, most of the participants said they had the tendency to imagine the presence of someone else inside the VR scenario. This highlights the possible contribution of imagination in recreating a realistic relaxing context if the elements also represent human beings. A higher sense of immersivity was experienced by the participants when they had the opportunity to personalize the VRE. This means that personalization elicited a greater sense of presence in the VR environment [57,64].

The participants experienced a greater degree of comfort after the personalized VR experience. Interestingly, according to their self-report, factors contributing to pleasantness and preference for the personalized VRE were as follows: (1) the correspondence of the personalized VR with the relaxing context they would have chosen in reality; (2) the reminiscence; (3) the possibility of choosing and controlling elements in the VR context; (4) the realism of the stimuli. Moreover, regarding users’ preferences for the personalized or standard VR scenarios, no differences emerged in our sample between male and female participants, both from a preliminary quantitative analysis (Chi-Square test = 0.56; *p* > 0.05) and a qualitative assessment.

Other studies have highlighted the usefulness of VR in promoting relaxation and reducing anxiety [18,29,65]. In line with these outcomes, our data suggest that exposure to a VR relaxing environment promoted the perception of a reduced state-anxiety symptoms and facilitated a sense of relaxation, independently of the possibility to personalize the VR context per se. Moreover, when the participants had the opportunity to personalize and choose the audio and visual elements to compose their favorite relaxing environment, they felt a more subjective state of relaxation compared to when they experienced the standard VRE. These findings on personalization are promising as they point out the importance of deploying a user-centered design approach [28,37] in adapting the VRE stimuli to fit the needs of individuals and to promote a positive attitude and a better engagement with the relaxation training. Personalization can also contribute to making virtual contexts more applicable than standard VRE. Future studies should shed further light on that. The investigation of the different types of relaxation training techniques (e.g., body scan, diaphragmatic breathing, PMRT), integrated into different types of VREs (e.g., personalized versus standard) should be further assessed in comparison with control group conditions.

Sometimes even well-established and effective training relaxation programs are not appreciated by the participants. For example, in the present study, two participants reported experiencing boredom and discomfort with the body scan technique, adding that the experience was engaging only thanks to the VRE. In other circumstances, they did not appreciate the relaxation training, considering it too dull. Thus, integrating exposure to relaxation training with VRE could also help to better fit users’ needs and support motivation to learn a valid relaxation technique that would not be experienced alone.

Even if this research and other studies have begun to investigate the combined role of VRE and relaxation training, so far findings are still preliminary in proving that a specific relaxation technique, either alone or in combination with a VRE, is more effective than others [18]. For this reason, randomized controlled studies should be conducted, comparing different experimental conditions with adequate sample sizes. Since not all of the relaxation training could be valid for everyone, it is important to consider individuals’ preferences and characteristics, by investigating personalization and promoting the validation of more specific and suitable treatment protocols based on the users’ needs. In addition, the relaxation training described in the present study implies an active role for the individual in promoting relaxation. Future studies may assess the role of coping strategies and the Locus of Control [66] in managing anxiety to understand if having an internal LoC could help to promote a better sense of relaxation by using the learning acquired during the procedure. Moreover, it would be interesting to assess if coping strategies change over time by comparing the assessment before and after the exposure.

Another relevant goal for future studies is to evaluate in more depth the positive emotions experienced during the procedure and investigate the impact of the vividness of visual imagery on the sense of presence perceived during the VR exposure [67].

This study has some limitations that affect the generalizability of our findings. The sample size is too small to provide strong conclusions. Experimental conditions were not compared with a control group, and this limited the significance of the results for the clinical practice. However, based on our aims, the findings of the current research are promising, but controlled and randomized trials should be conducted to investigate the efficacy of this type of digital relaxation intervention, including the deployment of a between-subject design approach to control intervention, including the deployment of possible “experimental demand” bias. It would be useful to stratify the participants’ sample according to socio-demographic variables, and to increase its size. Future studies may also include standardized questionnaires validated on the local target population

Finally, together with the effective self-report questionnaires administered in this work, future studies may also foresee the deployment of more objective physiological assessment methods, such as skin conductance and heart rate variability measures to assess relaxation and state-anxiety.

## 5. Conclusions

The study presented extends the state of the art research on the impact of personalized VREs, suggesting the role of personalization as a promising component in facilitating engagement, and promoting a higher perceived state of relaxation. These preliminary results contribute to the planning of future studies investigating the effect of relaxation training in personalized VREs on large-scale clinical and non-clinical samples. The efficacy and effectiveness of VRE for relaxation are likely to encourage the deployment of these digital interventions to promote mental well-being, making them more sustainable and scalable in clinical practice. Overall, the personalization of clinical interventions by the deployment of user-centered design methods helps consider people’s needs to improve their motivation and engagement, with positive implications for their well-being. Our exploratory outcomes are promising for implementing between-subject experimental design studies that should also be focused on deploying VR settings to improve and manage limits related to mental relaxing imagery techniques. This line of investigation could shed more light on the role of VR environments in favoring relaxation and autonomy, reducing costs and improving the deployment of treatments, particularly needed during pandemic conditions. Access to effective relaxation training protocols online may benefit the administration of psychological interventions, especially in situations where restrictions to face-to-face treatment are present. To conclude, this study’s results contribute to a better understanding of the user experience and preferences for the personalization of VREs for relaxation, paving the way to more extensive future analyses on the role of these solutions in supporting digital interventions for mental well-being.

## Figures and Tables

**Figure 1 ijerph-19-07237-f001:**
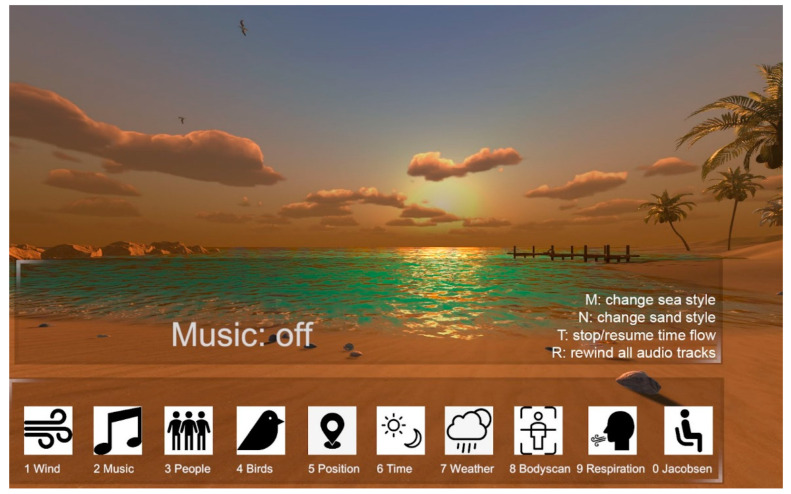
Screenshot of the interface.

**Figure 2 ijerph-19-07237-f002:**
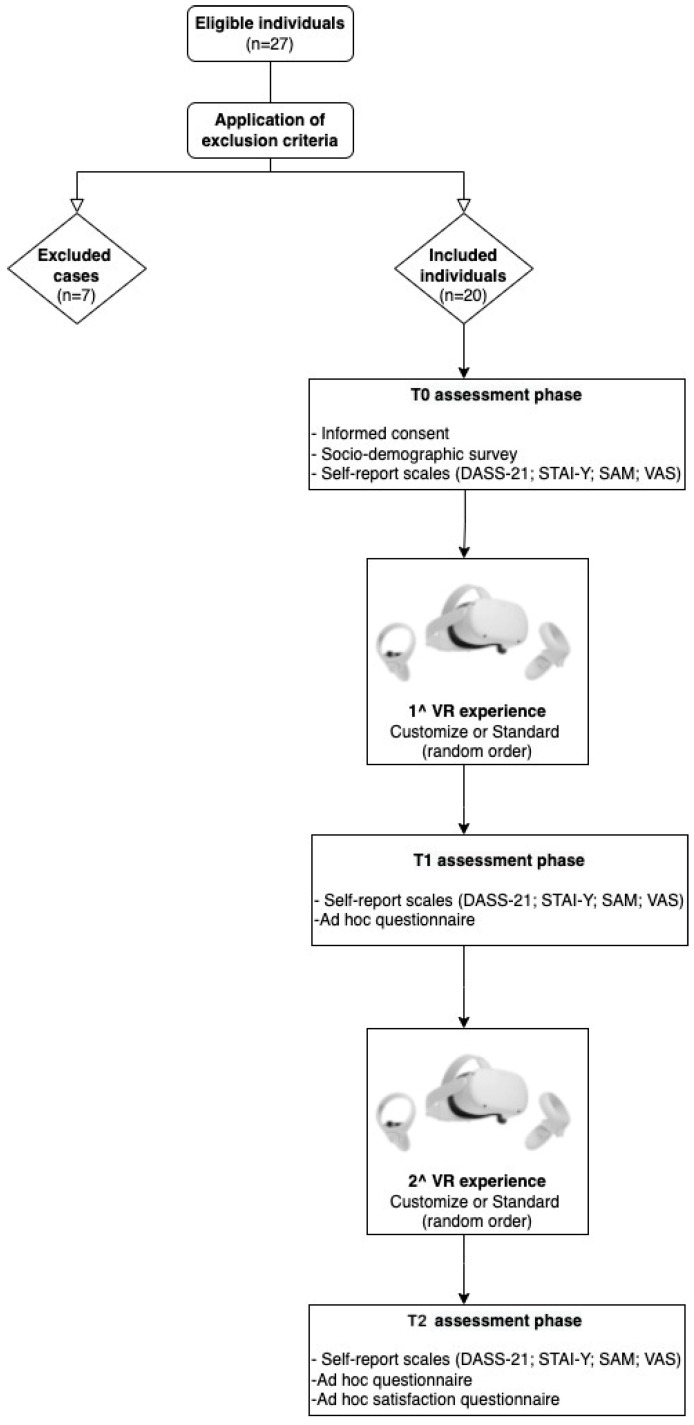
Flowchart procedure. Notes: DASS-21 = Depression, Anxiety and Stress Scale-21; STAI-Y = State-Trait Anxiety Inventory-Y1 and Y2; SAM = Self-Assessment Manikin; VAS = Visual Analogue Scale.

**Table 1 ijerph-19-07237-t001:** Data collection method and analysis.

Data collection Method	Analysis Captured
Socio-Demographic Survey	*Descriptive analysis.* Frequencies, means, and standard deviations related to socio-demographics variables.
Virtual Reality Symptom Questionnaire	*Descriptive analysis.* Frequencies related to cybersickness variables.
Questionnaire inspired by the Presence Questionnaire and the Immersive Tendencies Questionnaire	*Descriptive analysis.* Means and standard deviations. *ANOVAs repeated measures* to compare the self-report scores taken at two different time points.
Open-Ended Question–VR scenario preferred	*Descriptive analysis.* Thematic analysis, frequencies related to which VR scenario was preferred.
Ad Hoc Satisfaction Questionnaire	*Descriptive analysis.* Frequencies.
Depression, Anxiety and Stress Scale-21 State-Trait Anxiety Inventory-Y Self-Assessment Manikin Visual Analogue Scale	*ANOVAs repeated measures* and *Paired-samples t-tests*.

**Table 2 ijerph-19-07237-t002:** Comparison between randomization orders.

	Randomization Order Group	M (SD)	F (1,18)	*p*
**SAM_V_T1**	1 2	3.5 (1.1) 2.8 (1.5)	1.47	ns
**SAM_A_T1**	1 2	5.2 (2.4) 7.2 (1.9)	4.13	ns
**SAM_D_T1**	1 2	4.8 (2.2) 6.2 (1.7)	2.55	ns
**VAS_RELAX_T1**	1 2	7.8 (1.6) 8.2 (1.1)	0.43	ns
**VAS_PRESENCE_T1**	1 2	6.6 (2.4) 7.9 (2.1)	1.67	ns
**SAM_V_T2**	1 2	2.4 (0.8) 2.1 (1.3)	0.64	ns
**SAM_A_T2**	1 2	5.8 (2.7) 7.4 (2.1)	2.22	ns
**SAM_D_T2**	1 2	5.6 (1.7) 6.4 (1.7)	1.18	ns
**VAS_RELAX_T2**	1 2	7.6 (0.9) 8.4 (0.9)	2.95	ns
**VAS_PRESENCE_T2**	1 2	7.6 (0.9) 8.4 (0.9)	3.43	ns
**STAI-Y1_T1**	1 2	28.2 (7.6) 27.1 (4.5)	0.16	ns
**STAI-Y1_T2**	1 2	25.2 (3.9) 25.6 (4.7)	0.04	ns
**REALISM_T1**	1 2	100.2 (14.3) 103.2 (19.8)	0.15	ns
**REALISM_T2**	1 2	105.1 (17.5) 108.6 (12.9)	0.26	ns
**ENGAGEMENT_T1**	1 2	10.9 (2.2) 10.6 (2.9)	0.07	ns
**ENGAGEMENT_T2**	1 2	10.3 (2.9) 12.3 (1.4)	3.82	ns
**IMMERSIVITY_T1**	1 2	11.8 (5.5) 13.9 (3.1)	1.81	ns
**IMMERSIVITY_T2**	1 2	17.2 (1.5) 16.1 (2.4)	1.11	ns
**USABILITY_T1**	1 2	17.1 (3.1) 14.1 (3.7)	3.86	ns
**USABILITY_T2**	1 2	16.5 (3.5) 15.1 (3.4)	0.82	ns
**EMOTIONAL STATE_T1**	1 2	46.7 (7.5) 43.8 (8.4)	0.66	ns
**EMOTIONAL STATE_T2**	1 2	45.7 (7.1) 47.3 (6.1)	0.30	ns

Notes: Rand. order group = 1 (standard VRE before/personalized VRE after), 2 (personalized VRE before/standard VRE after); M (SD) = Mean (Standard Deviation); *p* = *p*-value; SAM_V = Self-Assessment Manikin_Valence; SAM_A = Self-Assessment Manikin_Arousal; SAM_D = Self-Assessment Manikin_Dominance/Control; VAS = Visual Analogue Scale; STAI-Y2 = State-Trait Anxiety Inventory-Y2 (regarding trait anxiety features); STAI-Y1 = State-Trait Anxiety Inventory-Y1 (regarding state-anxiety features).

**Table 3 ijerph-19-07237-t003:** Means and Standard Deviations related to self-report scales’ scores.

Questionnaire	M (SD)
**DASS_Tot**	9.9 (6.9)
**DASS_D**	2.5 (2.5)
**DASS_A**	1.7 (1.9)
**DASS_S**	5.7 (3.6)
**STAI_Y2**	34.9 (6.1)
**STAI_Y1 (T0)**	32.7 (9.1)
**STAI Y1 (st)**	27.7 (6.1)
**STAI Y1 (p)**	25.4 (4.2)
**SAM_V (T0)**	3.3 (0.7)
**SAM_V (st)**	3.2 (1.3)
**SAM_V (p)**	2.2 (1.1)
**SAM_A (T0)**	5.5 (1.7)
**SAM_A (st)**	6.2 (2.4)
**SAM_A (p)**	6.6 (2.5)
**SAM_D (T0)**	5.5 (1.4)
**SAM_D (st)**	5.5 (2.1)
**SAM_D (p)**	6 (1.7)
**VAS_Relax (T0)**	5.9 (1.6)
**VAS_Relax (st)**	8 (1.3)
**VAS_Relax (p)**	8.2 (1.6)
**VAS_Sense of Presence (st)**	7.3 (2.3)
**VAS_Sense of Presence (p)**	8 (1.1)
**Realism (st)**	101.7 (16.9)
**Realism (p)**	106.9 (15.1)
**Engagement (st)**	10.8 (2.6)
**Engagement (p)**	11.3 (2.5)
**Immersivity (st)**	12.9 (4.5)
**Immersivity (p)**	16.6 (2.1)
**Usability (st)**	15.6 (3.7)
**Usability (p)**	15.8 (3.4)
**Emotional_state (st)**	45.3 (7.9)
**Emotional state (p)**	46.5 (6.4)

Notes: T0 (assessment before VREs); st (assessment after standard VRE); p (assessment after personalized VRE); M (SD) = Mean (Standard Deviation); DASS_Tot = Depression, Anxiety and Stress Scale-21_Total score; DASS_D = Depression, Anxiety and Stress Scale-21_Depression scale; DASS_A = Depression, Anxiety and Stress Scale-21_Anxiety scale; DASS_S = Depression, Anxiety and Stress Scale-21_Stress scale; STAI-Y2 = State-Trait Anxiety Inventory-Y2 (regarding trait anxiety features); STAI-Y1 = State-Trait Anxiety Inventory-Y1 (regarding state-anxiety features); SAM_V = Self-Assessment Manikin_Valence; SAM_A = Self-Assessment Manikin_Arousal; SAM_D = Self-Assessment Manikin_Dominance/Control; VAS = Visual Analogue Scale.

**Table 4 ijerph-19-07237-t004:** Themes and quotes by users.

Main Theme	User Quotes
„Relaxing as in reality“	„Especially for the birdsong which is what I feel when I rest in the country“ (**Participant 2**). „I preferred the personalized environment because it reflects more of a context in which I relax in reality“ (**Participant 9**). „I find it easier to relax in the mountains“ (**Participant 10**). „Because, even in reality, when I have to relax I choose to go to the seaside“ (**Participant 11**). „The elements around me reflected an environment that generally makes me feel good, and relaxed“ (**Participant 8**). „If I want to take a break from everything, I usually go to the mountains“ (**Participant 3**). „I could select a context that generally relaxes me“ (**Participant 13**). „I usually feel more comfortable at the seaside, that’s why I chose it“ (**Participant 6**). „I prefer the seaside to the countryside in reality. I can associate it more with a sense of relaxation“ (**Participant 16**). „There were elements that helped me to relax as in everyday life“ (**Participant 5**). „This is very similar to my favourite place in reality“ (**Participant 7**). „It is similar to the place where I feel better about relaxing“ (**Participant 17**). „Because it represents a context that recalls the relaxing places in reality, and this was helpful“ (**Participant 18**). „I could recreate a more similar environment to the one that really relaxes me“ (**Participant 14**). „It was similar to the beach I choose to relax“ (**Participant 12**). „The seaside is one of the environment that I choose when I want to relax and disconnect my head“ (**Participant 19**).
„Reminiscence“	„The countryside scenario reminds me of the area from which I come. This makes me feel at home“ (**Participant 1**).
„Control“	„The very fact of having been able to choose“ (**Participant 15**). „I chose the mountain environment because I associated many beautiful memories with it“ (**Participant 9**). „Also because, as a child, I always went to the seaside and it was beautiful. This context somehow reminded me of it“ (**Participant 19**) „I decided what to insert in the environment. Then I liked it more!“ (**Participant 17**). „„First of all because I could decide“ (**Participant 19**). „It was my choice“ (**Participant 16**).
„Realism of the stimuli“	„In the standard context I found more elements similar to reality, I felt more comfortable“ (**Participant 20**). „The seaside seemed more like reality and, although it was not the environment I had chosen, in the end I preferred it“ (**Participant 4**).

**Table 5 ijerph-19-07237-t005:** Paired Samples *t*-test between psychological constructs administered at three different times.

	t (1,19)	*p*
**STAI Y1 (T0)/STAI Y1 (st)**	**3.20**	**<0.01**
**STAI Y1 (T0)/STAI Y1 (p)**	**4.97**	**<0.001**
**STAI Y1 (st)/STAI Y1 (p)**	**2.20**	**<0.05**
**SAM_V (T0)/SAM_V (st)**	0.42	ns
**SAM_V (T0)/SAM_V (p)**	**3.80**	**<0.01**
**SAM_V (st)/SAM_V (p)**	**2.97**	**<0.01**
**SAM_A (T0)/SAM_A (st)**	−1.58	ns
**SAM_A (T0)/SAM_A (p)**	**−** **2.46**	**<0.05**
**SAM_A (st)/SAM_A (p)**	−1.29	ns
**VAS_relax (T0)/VAS_relax (st)**	**−** **6.25**	**<0.001**
**VAS_relax (T0)/VAS_relax (p)**	**−** **2.52**	**<0.05**
**VAS_relax (st)/VAS_relax (p)**	**−** **9.95**	**<0.001**

Notes: T0 (assessment before VREs); st (assessment after standard VRE); p (assessment after personalized VRE); M (SD) = Mean (Standard Deviation); STAI-Y1 = State-Trait Anxiety Inventory-Y1 (regarding state-anxiety features); SAM_V = Self-Assessment Manikin_Valence; SAM_A = Self-Assessment Manikin_Arousal; SAM_D = Self-Assessment Manikin_Dominance/Control; VAS = Visual Analogue Scale; The bolded number correspond to data that are statistically significant.

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
