# Peer review of "The Role of Personalization in the User Experience, Preferences and Engagement with Virtual Reality Environments for Relaxation"

_ijerph, 2022, doi:10.3390/ijerph19127237_

Round 1

Reviewer 1 Report

This manuscript investigate the personalized virtual reality environment (VER) would considered as a more promising way in facilitating relaxation and promoting engagement. By further detailed describnation of how they set up the "personalized choice of the elements", analyzed the users experience and the author find out the personalized VER is more preferred compared with standard VER.

In this version, authors provided more detailed information about how to set up the personalized element during their study, also they include more detailed abstract, methods and interpretation of the resutls. I only have one minor question based on their findings:

the sample included in this study is female dominate (-80%), is there any gender preference on the personalized VER?

Author Response

Reviewer 1:

This manuscript investigate the personalized virtual reality environment (VER) would considered as a more promising way in facilitating relaxation and promoting engagement. By further detailed describnation of how they set up the "personalized choice of the elements", analyzed the users experience and the author find out the personalized VER is more preferred compared with standard VER. In this version, authors provided more detailed information about how to set up the personalized element during their study, also they include more detailed abstract, methods and interpretation of the resutls. I only have one minor question based on their findings:

The sample included in this study is female dominate (-80%), is there any gender preference on the personalized VER?

“Thank you for calling this to our attention. To the best of our knowledge, considering that this topic is innovative, there are no studies that have highlighted differences based on preference for a relaxing or standard virtual environment. Moreover, based on our aims, the study design implies that the sample size can be small because the focus is mainly on the user experience outcome, and sample size estimation is not required. Moreover, the sample can be selected from accessible individuals since this phase will aim to understand if the intervention deserves more deepened analysis, improvement, and future testing. As we have specified in the discussion section, future studies that intend to deeply investigate the efficacy of personalization could stratify the participants’ sample according to socio-demographic variables and increase its size. Anyway, considering other features and constructs that are investigated in the context of virtual reality, no gender differences has been put in evidence in postural tasks (Robert et al., 2016), in memory capacities employed during the simulation (Cárdenas-Delgado et al., 2017), in the experience of low- vs. high-quality simulation (Juan et al., 2018), in cognitive abilities as information recognition and retrieval as well as a cognitive load during the simulation (Roettl & Terlutter, 2018), and in the sense of presence and performance (Khashe et al., 2018). Though other studies that investigated the sense of presence highlighted those male participants frequently reported a higher sense of spatial presence, more perceived realism, and higher levels of the sense of being in the environment than female participants (Felnhofer et al., 2012). Finally, Rangelova and Marsden (2018) found no significant gender differences in enjoyment and interest in the simulation.

Based on preliminary quantitative data, no differences emerged in our sample considering gender preference (Chi-Square test= 0.56; p>.05).). Moreover, from a qualitative point of view, no differences emerged between males and females in the preferences expressed in the personalized and standard VR scenarios.

We have added this point in the discussion section (Lines 1680-1683) as follows:

“Moreover, regarding users preferences for the personalized or standard VR scenarios, no differences emerged in our sample between males and females participants, both from a preliminary quantitative analysis (Chi-Square test= 0.56; p>0.005) and qualitative assessment.”

References:

  • Robert, M. T., Ballaz, L., & Lemay, M. (2016). The effect of viewing a virtual environment through a head-mounted display on balance. Gait Posture 48, 261–266. doi: 10.1016/j.gaitpost.2016.06.010

  • Cárdenas-Delgado, S., Méndez-López, M., Lizandra, M. D. C. J., Pérez-Hernández, E., Lluch, J., & Vivó, R. (2017). Using a virtual maze task to assess spatial short-term memory in adults. VISIGRAPP 1: GRAPP. 46–57. doi: 10.5220/0006093200460057

  • Juan, M., García-García, I., Mollá, R., & López, R. (2018). Users' perceptions using low-end and high-end mobile-rendered HMDs: a comparative study. Computers 7:15. doi: 3390/computers7010015

  • Roettl, J., & Terlutter, R. (2018). The same video game in 2D, 3D or virtual reality–How does technology impact game evaluation and brand placements? PLoS ONE 13:e0200724. doi: 10.1371/journal.pone.0200724

  • Khashe, S., Becerik-Gerber, B., Lucas, G., & Gratch, J. (2018). “Persuasive effects of immersion in virtual environments for measuring pro-environmental behaviors,” in ISARC. Proceedings of the International Symposium on Automation and Robotics in Construction, 35. (Waterloo, ON: IAARC Publications), 1–7. doi: 10.22260/ISARC2018/0167

  • Felnhofer, A., Kothgassner, O.D., Beutl, L., Hlavacs, H., & Kryspin-Exner, I. (2012). Is Virtual Reality made for Men only? Exploring Gender Differences in the Sense of Presence.

  • Rangelova, S., & Marsden, N. (2018). Gender Differences Affect Enjoyment in HMD Virtual Reality Simulation. Available Online at: https://www.researchgate.net/profile/Stanislava_Rangelova/publication/327534834_Gender_Differences_Affect_Enjoyment_in_HMD_Virtual_Reality_Simulation/links/5b93d36892851c78c4fcd7e7/Gender-Differences-Affect-Enjoyment-in-HMD-Virtual-Reality-Simulation.pdf (accessed on March 26 2020).

  • Grassini, & Laumann, K. (2020). Are Modern Head-Mounted Displays Sexist? A Systematic Review on Gender Differences in HMD-Mediated Virtual Reality. Frontiers in Psychology, 11.  

Doi: 10.3389/fpsyg.2020.01604

Reviewer 2 Report

The authors state, that 7 patients were excluded from the tests because of the following criteria:

i) participant had a mental disorder diagnosis, ii) a score on the STAI-Y2 indicating moderate or severe trait-anxiety levels (raw cut point >50; [46]), iii) a score > 9 on the DASS-21 depression subscale, iv) a score > 6 on the DASS-21 anxiety scale, v) a score > 10 on the DASS stress scale.

What was the reason for such exclusion? The paper should be extended with proper explanation.

In line 406 the author state:

"Eighteen of the individuals (90%) preferred the personalized environment. Of these,16/20 (89%) said"

This seems contradictory as the second sentence refers to those 18 individuals (and percentage shows that) where the absolute numbers refers to total number of participants.

The Figure 2 description on page 7 is separate from the figure itself.

In general the paper is sound however I am missing in discussion a part where the level of relaxation was clearly compared (even subjectively). The question is whether the additional setup required, more configurations etc. Is it really worth it? Such discussion would in my opinion increase the value of the paper.

Author Response

Reviewer 2:

1) The authors state, that 7 patients were excluded from the tests because of the following criteria:

  1. i) participant had a mental disorder diagnosis, ii) a score on the STAI-Y2 indicating moderate or severe trait-anxiety levels (raw cut point >50; [46]), iii) a score > 9 on the DASS-21 depression subscale, iv) a score > 6 on the DASS-21 anxiety scale, v) a score > 10 on the DASS stress scale.What was the reason for such exclusion? The paper should be extended with proper explanation.

Thank you for this comment. Ours was a convenience sample of people extracted from a non-clinical population. For this reason, and to control the effect that mental disorders diagnosis, trait-anxiety and depressive symptomatology could have when comparing the state-anxiety condition before and after VREs, seven participants were excluded from the study.

To clarify this information, the procedure section has been modified as follows:

(Lines 392-394): “Seven participants were excluded to control the effect that mental disorders diagnosis, trait-anxiety and depressive symptomatology may have when comparing state-anxiety before and after VREs usage.

2) In line 406 the author state:

"Eighteen of the individuals (90%) preferred the personalized environment. Of these,16/20 (89%) said" This seems contradictory as the second sentence refers to those 18 individuals (and percentage shows that) where the absolute numbers refers to total number of participants.

Thank you for the comment. The sentence has been modified as follows (Lines 1385-1388):

“Of these 18 users, 16 (89%) said it represented the real context they would choose to relax, and they found very important the possibility to personalize the elements to make the VRE more likely not only to the preferred context but to the place in the real world where they would go to relax.”

3) The Figure 2 description on page 7 is separate from the figure itself.

Thank you for the comment. The correction has been made.

4) In general the paper is sound however I am missing in discussion a part where the level of relaxation was clearly compared (even subjectively). The question is whether the additional setup required, more configurations etc. Is it really worth it? Such discussion would in my opinion increase the value of the paper.

Thank you for the suggestion. About this point in the discussion section we said that (Lines 1669-1755):

Moreover, when participants had the opportunity to personalize and choose the audio and visual elements to compose their favourite relaxing environment, they felt a more subjective state of relaxation compared to when they had experienced the standard VRE. These findings on personalization are promising as they point out the importance of deploying a user-centered design approach [28, 37] in adapting the VRE stimuli to fit the needs of individuals and to promote a positive attitude and a better engagement with the relaxation training. Personalization can also contribute to make virtual contexts more applicable than standard VRE. Future studies should shed further light on that. The investigation of the different types of relaxation training techniques (e.g. body scan, diaphragmatic breathing, PMRT), integrated into different types of VREs (e.g. personalized versus standard) should be further assessed in comparison with control group conditions.”

This is a promising result that future research could further investigate.

Moreover, it is important to consider that the present research is an exploratory study. We aimed to investigate the user experience to obtain preliminary data about preferences, pleasure, satisfaction, engagement, immersivity, sense of presence and the subjective perception of relaxation and state-anxiety whether being exposed to a standard and a personalized relaxing VREs. It was not focused on investigating the efficacy of this kind of intervention for clinical deployment.

To this aim and based on the consideration that emerged from the previous report, we have deeply modified the structure of our study to highlight the qualitative outcomes collected, suggesting that VR scenarios are favoring relaxation. Moreover, by following this request, we have highlighted the limits and future work needed to shed further light on this topic.

Reviewer 3 Report

Studying VR users' experiences can be worthwhile.. However, there are many things that need to be supplemented in writing the manuscript, and the necessity of research or the validity of the results of your study must be convincingly presented.

These are the things that need to be supplemented:

1. Abstract needs to be summarized to make it more readable.

2. Overall, the paragraph is too short. There are a lot of paragraphs where you only give a little bit of information. If possible, please avoid one-sentence paragraphs in the manuscript.

3. Describe the experimental procedure and explain why you did so. For example, you need to elaborate on what controls the experiment to ensure the internal validity of the results of this study.

4. It is recommended that you describe a scale in one paragraph.

5. When you are presenting statistics or numbers, I hope you meet the guideline for submission of manuscript or international standards. For example, if you have a number greater than 1, you must mark 0 before the decimal point. And please treat decimal points with dot, not comma, according to international standard.

6. I want you to follow scientific qualitative research methods when presenting qualitative analysis results.

7. It seems that you could have increased the number of samples for like this experiment study to make statistical analysis easier and generalizable. In fact, there doesn't seem to be much to conclude with this kind of experiment. It just seems like exploratory study. So you have to present the value of research convincingly enough to quell these doubts.

Author Response

Reviewer 3:

Studying VR users' experiences can be worthwhile.. However, there are many things that need to be supplemented in writing the manuscript, and the necessity of research or the validity of the results of your study must be convincingly presented.

  1. Abstract needs to be summarized to make it more readable.

Thank you for the comment. We added information about the procedure following the indication of the previous report. Otherwise, we have provided a more comprehensive version of the abstract, as follows (Lines 17-35):

“Abstract: Virtual Reality Environments (VREs) are widely deployed in mental health treatments, often associated with relaxation techniques. Personalization of natural VR-based scenarios is a key element that can further facilitate users' sense of presence and relaxation. This study explored the role of VREs personalization on the user experience with an environment supporting relaxation, by deploying mixed methods.  Methods: A non-clinical sample of 20 individuals exposed to a VRE supporting body scan guided relaxation participated. In the personalized condition, participants had the option to choose the context (e.g. sea, mountain, or countryside) and include in the scenario different types of sounds, visual elements, change the time of day and weather. In the standard condition, individuals were exposed to a relaxing VRE, but they could choose neither the context nor the auditory and visual elements. The order of presentation of the personalized vs non-personalized environments was randomized. Measures regarding relaxation, state-anxiety perceived levels, VRE-related symptoms, the usability of the VR setting, sense of presence, pleasure, activation, engagement, and level of immersion experienced were collected before and after exposure to the VR environments. Results: Findings showed that personalized VREs were preferred by users. Participants generally referred to experience a greater immersivity, pleasure, engagement, and relaxation in the personalized virtual settings. Conclusion: The study further confirms the role of personalization as a component positively contributing to relaxation and engagement. Future research may further assess this effect in the context of large-scale controlled studies involving clinical and non-clinical populations.”

  1. Overall, the paragraph is too short. There are a lot of paragraphs where you only give a little bit of information. If possible, please avoid one-sentence paragraphs in the manuscript.

Thank you for the suggestions. We have modified throughout the text based on this indication.

  1. Describe the experimental procedure and explain why you did so. For example, you need to elaborate on what controls the experiment to ensure the internal validity of the results of this study.

Thank you a lot for this comment. In general, we can affirm that results obtained are related to a careful choice of the type of sample, the materials used, and the correct conduct of statistical analysis based on the aims and the experimental design. We described in more detail the experimental design, explaining why we took certain methodological and analytical choices. We added the following information:

(Lines 236-302): “The study of user experience related to the role of personalized VR scenarios has attracted interest in the scientific community although, as mentioned before, the studies published are more on a theoretical level [e.g. 37]. To further understand the role of personalization component in VR scenarios, the main aim of our pilot and exploratory study was to investigate the experience of users to obtain preliminary data about preferences, pleasure, satisfaction, engagement, immersivity, sense of presence and the subjective perception of relaxation and state-anxiety whether being exposed to a standard and a personalized relaxing VREs. To this end, inspired by the user experience’s background described above, and to generate integrated support upon qualitative and quantitative data, we obtained information about the physical discomfort, related to the heaviness or encumbrance of the head-mounted display, the simulator-sickness, the wayfinding, and the experience with the auditory and visual elements, both in the standard and personalized scenarios. An explicit emphasis has been placed on engagement, immersivity, sense of presence, subjective perception of relaxation and state-anxiety levels, and pleasure. The study design was based on a mixed methods approach inspired by the Obesity-Related Behavioral Intervention Trials (ORBIT) framework [46] for design (Phase Ib) of digital interventions and their preliminary testing (Phase IIa). In this phase, a proof-of-concept implementation of the personalized VR scenario was realized, and preliminary testing was done to investigate level of engagement and the user experience with a convenience sample of non-clinical individuals. In this context, it is useful to apply a within-subjects design where users act as their controls in a pre-post comparison. The sample size can be small because the focus is mainly on the user experience outcome, and sample size estimation is not required. Moreover, the sample can be selected from accessible individuals since this phase will aim to understand if the intervention deserves more deepened analysis, improvement, and future testing. We hypothesized that: 1) relaxing VR scenarios promote relaxation and reduce state-anxiety symptoms, 2) the possibility of personalizing the VR environment can be perceived as more engaging for users. By following a person-centered approach to enhancing the effect of standard interventions [28], we present below the main outcomes of our design and preliminary testing phases.”

  1. It is recommended that you describe a scale in one paragraph.

Thank you for the suggestions. We have modified the Materials section based on this indication.

  1. When you are presenting statistics or numbers, I hope you meet the guideline for submission of manuscript or international standards. For example, if you have a number greater than 1, you must mark 0 before the decimal point. And please treat decimal points with dot, not comma, according to international standard.

Thank you for the comment. All statistics or numbers presented in the manuscript are revised based on the “APA style 7th edition – Numbers and Statistics Guide”.

  1. I want you to follow scientific qualitative research methods when presenting qualitative analysis results.

Based on this consideration, in the results section, we have described the method we used for extrapolating themes from the open-ended question about the preferred VR scenario, the data collection method and their analysis (Lines 650-656) as follows:

“Descriptive data about user experience, preferences, and engagement were derived from frequencies. Moreover, participants’ responses to the open-ended question related to the preferred VR scenario were analyzed by SP and SC by using thematic analysis and were reported as frequencies. Data were analyzed thematically following an inductive, data-driven approach based on the procedure outlined by Braun and Clarke [63]. Data codes were generated systematically, then collated into themes and applied to the entire data set to generate frequencies.”

Moreover, we have modified the 3.4.5. paragraph and added in Table 4 (1385-1386) all the users quote those participants said about their preference.

In general, all the methods and results section have been revised to explain better results.

  1. It seems that you could have increased the number of samples for like this experiment study to make statistical analysis easier and generalizable. In fact, there doesn't seem to be much to conclude with this kind of experiment. It just seems like exploratory study. So you have to present the value of research convincingly enough to quell these doubts.

Considering that the present research is an exploratory study, our aim was to investigate the user experience to collect preliminary data about preferences, pleasure, satisfaction, engagement, immersivity, sense of presence and the subjective perception of relaxation and state-anxiety whether being exposed to a standard and a personalized relaxing VREs. The study was not mainly focused on investigating the efficacy of this kind of intervention for clinical deployment.  

We have deeply modified the presentation of our study to highlight the qualitative outcomes, suggesting that VR scenarios can facilitate relaxation. Moreover, we have highlighted the limits and future work needed to shed further light on this topic.

All the manuscript sections have been modified accordingly.

Round 2

Reviewer 3 Report

Thank you for the revision according to my suggestions or opinions. I think the quality of the manuscript has increased, and the readability has also increased. 

I evaluate that it can be published in this journal if the authors review thoroughly your manuscript and improve the quality of the paper. 

Thank you again for your efforts.

This manuscript is a resubmission of an earlier submission. The following is a list of the peer review reports and author responses from that submission.

Round 1

Reviewer 1 Report

The article is interesting but it may be interesting if the authors improve the following points:
- Describe your method better.
- Describe your results better.
- Develop the conclusion a bit more by giving research perspectives.
- Check the language.

Author Response

Reviewer 1

The article is interesting but it may be interesting if the authors improve the following points:

- Describe your method better.

Thank you for the comment. We have integrated the “Methods and Materials” section with helpful information to understand the level of personalization better, as follows:

Lines 128-141: “Specifically, users may choose to experience relaxation, based on their preference, in one of three realistic scenarios, a mountain, a marine, and a countryside environment, that may be selected and personalized with the support of a technical operator.

For each selected environment, participants had the option to personalize a series of sub-categories of elements, including different types of sounds (e.g. animal sounds, wind rustle, type of music), visual elements (e.g. the presence of people, objects or animals), the position into the virtual space, the presence of people, and change the time of day and weather conditions (an example has been represented in Figure 1). The operator had the possibility to customize each scenario based on a dashboard interface connected to Oculus via cable. Specifically, the interface presented a series of icons related to the different customizable variables (such as music, wind, weather conditions, time of day, presence of people) (Figure 1). The connection between PC and viewer allowed the operator to see what the participant observed during the experience.”

- Describe your results better.

Thank you for the suggestion. To simplify the reading of the results, we have modified the 3.4 paragraph as follows:

Lines 338-356: “Paired-samples t-test were conducted as a post-hoc analysis to understand the origins of differences between the three assessment time points. As shown from the STAI-Y1 and the VAS relax scale (Table 3), state-anxiety symptoms were higher before both the standard and personalized VR experiences. Indeed, individuals expressed, on average, a lower feeling of relaxation and more intense activation and anxiety levels before the exposure to both the VR scenarios (t STAI (T0) / STAI (st) = 3,20; p<0.01; t STAI (T0) / STAI (p) = 4,97; p<0.001; t VAS_rel (T0) / VAS_rel (st) = -6,25; p<0.001; t VAS_rel (T0) / VAS_rel (p) = -2,52; t SAM_A (T0) / SAM_A (p) = -2,46; p<0.05). Moreover, higher relaxation levels have been experienced after the personalized VR experience rather than the standard one (t STAI (st) / STAI (p) = 2,20; p<0.05; t VAS_rel (st) / VAS_rel (p) = -9,95; p<0.001). Re-garding the Valence of emotions, participants referred to have experienced a greater degree of pleasantness after the personalized VR setting (t SAM_V (T0) / SAM_V (p) = 3,80; p<0.01; t SAM_V (st) / SAM_V (p) = 2,97; p<0.01).

Overall, a higher impact of the personalized VR scenario vs the standard one on relaxation and anxiety management was found. These data provide initial evidence that the personalized scenarios work better than the standard ones.”

- Develop the conclusion a bit more by giving research perspectives.

Thank you for the suggestions. We have integrated the Conclusions section as follows:

Lines 499-510: “Overall, personalization of clinical interventions by the deployment of user-centred design methods helps to take into account people’s needs to improve their motivation and engagement, with positive implications for their well-being. Further studies may consider other relaxation protocols (such as PMRT and deep breathing) and comparing their effect in association with personalized VR relaxing scenarios. Moreover, between-subject experimental design approaches and the deployment of VR settings to improve mental imagery techniques could shed more light on the role of VR environments in favouring relaxation and autonomy, reducing the costs of treatments, particularly needed during pandemic conditions. The delivery of effective relaxation training protocols online may benefit the administration of psychological interventions, especially in situations where face-to-face treatment cannot be delivered.”

- Check the language.

A native English teacher also revised the English language. All the corrections are tracked in the manuscript.

Reviewer 2 Report

This study aims to investigate if personalized virtual reality environment (VRE) would work better (in terms of relaxation) than non-personalized VRE when combined with body scan technique. The authors found that the results, based on subjective feedback from the users that the personalized version worked better. I like the idea of personalization, though I have serious concerns about the study design, details below.

The idea of personalized VR is introduced somewhere around Line 81 and 82, here I suggest the authors to give concrete examples of what it means to “personalize”—is it different types of sceneries depending on each user’s preference (e.g., mountain vs. beach)? Or is it the same scenario (e.g., mountain), but with subtle sub-category personalization (e.g., river or no river). Some context like these would help the readers understand what level of personalization the authors are talking about here.

Line 100 to 108 mentions reference 18, not sure what the relevance is here since that study doesn’t have personalization, right? Also, did that study include a no-VR condition? Otherwise how do we know VR was helpful at all? Could simply be the effect of body scan over controlled breathing.

The parameters being personalized in this study (music, wind, weather conditions, time of day, presence of people, etc) should be included in abstract

Major concern: The authors opted for a within-subject design, so that all subjects went through the personalized vs. non-personalized VR program. This is a major flaw of the design as it is prone to participants’ own biases since the purpose of the experiment has become obvious to the participants (at least during T2). As such, we now do not know if the participants truly feel that the personalized program is indeed better, or they are simply playing along, this is something known as the “characteristic demand” or “experimental demand” in psychology and needs to be controlled for. Randomized order cannot solve this since participants already know what scores they gave in T1, so whatever condition they get in T2 they can adjust scores accordingly. I can only think of 3 ways to address this:

  • A between-subject design would solve this if the authors can use the existing T1 data (discard T2 since that is contaminated), and then collect new T1 for another group so that there is equal number between personalized vs. non-personalized data.
  • Add an objective/physiological component such as skin conductance or heart rate so that one can verify whether the participants truly, objectively, experience better outcome in the personalized condition. Right now everything is questionnaires, which would have been okay if a between-subject design is used.
  • No additional data collection, but rewrite the current manuscript substantially to only focus on participants’ subjective preference, because we don’t know if personalization really is objectively better than non-personalized VR from the current data. That is, the entire result of this study can be experimental artifact due to the within-subject design. Therefore, the authors would be constrained to talk about users’ “preferences” (i.e., people prefer personalized option over standard), but not how “effective” the personalization actually is (i.e., the personalized option actually works better than standard).

Minor comment:

Figure 2: T1 at the bottom should be T2

Author Response

Reviewer 2

This study aims to investigate if personalized virtual reality environment (VRE) would work better (in terms of relaxation) than non-personalized VRE when combined with body scan technique. The authors found that the results, based on subjective feedback from the users that the personalized version worked better. I like the idea of personalization, though I have serious concerns about the study design, details below.

  • The idea of personalized VR is introduced somewhere around Line 81 and 82, here I suggest the authors to give concrete examples of what it means to “personalize”—is it different types of sceneries depending on each user’s preference (e.g., mountain vs. beach)? Or is it the same scenario (e.g., mountain), but with subtle sub-category personalization (e.g., river or no river). Some context like these would help the readers understand what level of personalization the authors are talking about here.

Thank you for the comment. About the VR scenarios described in the present study, we introduced as follows from Lines 128-141:

“Specifically, users may choose to experience relaxation, based on their preference, in one of three realistic scenarios, a mountain, a marine, and a countryside environment, that may be selected and personalized with the support of a technical operator. For each selected environment, participants had the option to personalize a series of sub-categories of elements, including different types of sounds (e.g. animal sounds, wind rustle, type of music), visual elements (e.g. the presence of people, objects or animals), the position into the virtual space, the presence of people, and change the time of day and weather conditions (an example has been represented in Figure 1). The operator had the possibility to customize each scenario based on a dashboard interface connected to Oculus via cable. Specifically, the interface presented a series of icons related to the different customizable variables (such as music, wind, weather conditions, time of day, presence of people) (Figure 1). The connection between PC and viewer allowed the operator to see what the participant observed during the experience.”

  • Line 100 to 108 mentions reference 18, not sure what the relevance is here since that study doesn’t have personalization, right? Also, did that study include a no-VR condition? Otherwise how do we know VR was helpful at all? Could simply be the effect of body scan over controlled breathing.

We appreciate your consideration. On Line 126, we introduced Pizzoli et al. (2019) to cite studies investigating the effect of relaxation training integration in VREs (see also Lines 123-125).

About Line 110, we thank you for the comment since reference 18 could not be appropriate here. We modified it as follows:

Lines 105-114: “However, few studies so far have investigated in depth the role of personalization of VR-based scenarios as an element that may promote relaxation [37], and the effect of the integration of relaxation training in VREs. One of the most recent studies in this direction [18] had shown that being exposed to a virtual environment for relaxation including a body scan guiding audio track was more effective in improving participants’ relaxation if compared to VREs including a respiration control audio track. To further investigate the personalization component, the main aim of our study was to assess whether being exposed to a personalized VR relaxing environment could better facilitate user relaxation, engagement, satisfaction, pleasure, immersivity, and sense of presence compared to a non-personalized setting.”

  • The parameters being personalized in this study (music, wind, weather conditions, time of day, presence of people, etc) should be included in abstract.

Thank you for the suggestion. We have introduced this information in the abstract section as follows:

Lines 22-24: “Specifically, participants had the option to include different types of sounds (e.g. animal sounds, wind rustle, type of music), visual elements (e.g. the presence of people, objects or animals), change the time of day and weather conditions.”

  • Major concern: The authors opted for a within-subject design so that all subjects went through the personalized vs. non-personalized VR program. This is a major flaw of the design as it is prone to participants’ own biases since the purpose of the experiment has become obvious to the participants (at least during T2). As such, we now do not know if the participants truly feel that the personalized program is indeed better, or they are simply playing along, this is something known as the “characteristic demand” or “experimental demand” in psychology and needs to be controlled for. Randomized order cannot solve this since participants already know what scores they gave in T1, so whatever condition they get in T2 they can adjust scores accordingly. I can only think of 3 ways to address this:
  • A between-subject design would solve this if the authors can use the existing T1 data (discard T2 since that is contaminated), and then collect new T1 for another group so that there is equal number between personalized vs. non-personalized data.
  • Add an objective/physiological component such as skin conductance or heart rate so that one can verify whether the participants truly, objectively, experience better outcome in the personalized condition. Right now everything is questionnaires, which would have been okay if a between-subject design is used.
  • No additional data collection, but rewrite the current manuscript substantially to only focus on participants’ subjective preference, because we don’t know if personalization really is objectively better than non-personalized VR from the current data. That is, the entire result of this study can be experimental artifact due to the within-subject design. Therefore, the authors would be constrained to talk about users’ “preferences” (i.e., people prefer personalized option over standard), but not how “effective” the personalization actually is (i.e., the personalized option actually works better than standard).

We are grateful for these comments. To try to manage these points, we have considered to rephrase and integrate the discussion section as follows:

Lines 412-415: “Moreover, it investigated whether participants preferred the personalized scenario over the standard one and the role of a personalized virtual context in increasing relaxation and comfort as well.”

Lines 482-485: “Moreover, the within-subject design is limited by the possibility that participants realized the aim of the research, affecting their responses. For this reason, future studies may consider a between-subject design approach to investigate the effect of personalized VR settings to better control the possible “experimental demand” bias.

Another limitation of this study concerns the assessment measures used that were mainly based on self-report questionnaires. To verify whether participants objectively experience better outcomes in the personalized VR scenarios, future studies may include the collection of physiological measures (e.g. skin conductance and heart rate variability) to more objectively assess the well-being conditions of participants.”

Minor comment:

Figure 2: T1 at the bottom should be T2

Thank you for the correction. We have modified Figure 2 (Line 219).

Reviewer 3 Report

In this manuscript, authors studied the personalized virtual reality environment in supporting with body can which could pacilitate the relaxation and better manage anxiety. This manuscript has very clear background introduction, study design and detailed methodology about the procedure and evaluation.

Overall, this manuscript provide some evidence about the personalized virtual reality environment could be further exploited in future clinical assessment. However, there are some limitations in this study.

  1. the sample size is low (=20) to compare the standard or personalized virtual reality environment in together with body scan;
  2. also this study only include the non-clinical sample, which limited the sginificance in the clinical practice.

Author Response

Reviewer 3

In this manuscript, authors studied the personalized virtual reality environment in supporting with body can which could pacilitate the relaxation and better manage anxiety. This manuscript has very clear background introduction, study design and detailed methodology about the procedure and evaluation.

Overall, this manuscript provide some evidence about the personalized virtual reality environment could be further exploited in future clinical assessment. However, there are some limitations in this study.

  1. the sample size is low (=20) to compare the standard or personalized virtual reality environment in together with body scan;
  2. also this study only include the non-clinical sample, which limited the sginificance in the clinical practice.

Thank you for your suggestions. We have integrated the discussion section as follow:

Lines 470-473: “This study has some limitations that affect the generalizability of our findings. The sample size is small to provide strong conclusions. Experimental conditions were not compared with a control group, and this limited the significance of results for the clinical practice.”

Round 2

Reviewer 2 Report

I appreciate the authors' responses, although I stand by my previous review and do not think my concern can be addressed with a limitation statement. I should withdraw from this review process, thanks.